# Does *Bacillus thuringiensis* Affect the Stress and Immune Responses of *Rhynchophorus ferrugineus* Larvae, Females, and Males in the Same Way?

**DOI:** 10.3390/insects13050437

**Published:** 2022-05-06

**Authors:** Monica Celi, Debora Russo, Mirella Vazzana, Vincenzo Arizza, Barbara Manachini

**Affiliations:** 1Department of Biological, Chemical and Pharmaceutical Sciences and Technologies (STEBICEF), University of Palermo, Via Archirafi, 18-90123 Palermo, Italy; monica.celi@unipa.it (M.C.); debora.russo@unipa.it (D.R.); mirella.vazzana@unipa.it (M.V.); vincenzo.arizza@unipa.it (V.A.); 2Department of Agricultural, Food and Forest Sciences (SAAF), University of Palermo, V. le delle Scienze, 13-90128 Palermo, Italy

**Keywords:** median lethal dose, median lethal time, stress response, hemocytes, brain, Hsp70

## Abstract

**Simple Summary:**

*Rhynchophorus ferrugineus* is a destructive quarantine pest of palm trees, now widely distributed. Although broad-spectrum insecticides are often used to protect palm against *R. ferrugineus*, there is increasing concern about their effects on the environment and human health, especially where palm trees are located in urban areas. As an environmentally friendly entomopathogen, *Bacillus thuringiensis* (Bt) has been widely used to prevent other pest infestations. Although Bt products are the most sold bio-insecticides, there are still many interesting features to be investigated in the relationship of Bt and its hosts. We investigated the effect of Bt on larvae, females, and males. This research yielded experimental evidence of significant mortality and significant effects on immune system and stress answer. Within a few hours, stress due to Bt infection was detected in the hemocytes and in the brain providing better insights into the insect-pathogen interaction and highlighting the potential use of Bt in *R. ferrugineus* management.

**Abstract:**

*Bacillus thuringiensis* (Bt) is considered a potentially useful entomopathogen against red palm weevil (RPW) *Rhynchophorus ferrugineus*. We compared the effects of Bt on mature larvae, females, and males. The pathogenicity of Bt was evaluated, estimating: Median Lethal Dose (LD_50_), Median Lethal Time (LT_50_), Total Hemocyte Count (THC), and Differential Hemocyte Counts (DHC), and the expression of the stress protein Heat Shock Protein 70 (Hsp 70) in hemocytes and the brain. Mortality exhibited a positive trend with the dosage and duration of exposure to Bt. Larvae were more susceptible than adults, and the LD_50_ of females was almost double the value of that of the larvae. LT_50_ value was higher for females than for males and larvae. Treatment with sub-lethal doses of Bt induced a decrease in THC in larvae, females, and males. In treated larvae, plasmatocytes decreased, while oenocytes and spherulocytes increased. In treated females, all types of hemocytes decreased, while in males the number of plasmatocytes decreased and granulocytes increased. We also registered the stress response directly on hemocytes showing that, already at 3 h after eating Bt, the expression of the stress protein Hsp 70 was modulated. This effect was also observed in brain tissue at 6 h after treatment. The results confirm that Bt treatment induces a pathogenic state in larvae and adults of both sexes, with effects after only a few hours from ingestion; however, the effects are different in magnitude and in type of target.

## 1. Introduction

The use of specific entomopathogenic bacteria, such as *Bacillus thuringiensis* (Berliner, 1915) (Bacillales: Bacillaceae) (Bt), to target pests offers an ecologically sound and effective solution to pest problems. These bacteria pose less of a threat to the environment and to human health, and their use is increasing, particularly for controlling urban insect pests or invasive alien species. Insects are continually exposed to microorganisms, and thus, they have evolved effective immune defense systems to eliminate pathogens, including humoral and cellular defenses [1,2]. Studying the immune system of insects is important because it can lead to useful knowledge for the improvement of biological pesticides and increase the knowledge regarding host-pathogen relationships; thus, the red palm weevil (RPW), *Rhynchophorus ferrugineus* (Olivier, 1790) (Coleoptera: Dryophthorinae), has been proposed as a useful model for studying certain aspects of host-pathogen interactions [3,4,5,6,7] and insect innate immunity [8,9,10].

Several pathogens belonging to the *Bacillus* genus (*B. sphaericus*, *B. megaterium*, and *B. laterosporus*) are known to infect the larvae of this weevil, including *B. thuringiensis* [4,11,12]. Moreover, bacteria isolated from pupae were also affiliated with *Bacillus* [13]. Bt is a Gram-positive spore-forming bacterium, and most studies have focused on specific insecticidal toxins produced by *Bacillus* such as Cry and Cyt toxins (insecticidal proteins produced by *B. thuringiensis* during the sporulation phase as parasporal crystals (Cry) that have hemolytic and cytolytic activity (Cyt)) [14]. However, it has been shown that the death of insects after infection with *B. thuringiensis* is caused by the bacterial invasion of the hemocel, causing septicemia [15]. *R*. *ferrugineus* is considered one of the major pests of palm trees, especially Arecaceae. In Europe, it mainly attacks *Phoenix canariensis* Hort. Ex Chabaud, but several other species are also severely infested such as *Washingtonia filifera*, (Rafarin), *W. robusta* H. Wendl., *P. dactylifera* L., and *Chamaerops humilis* L., an endemic species [12,16]. All different instars of RPW (eggs, larvae, pupae, and adults) are present simultaneously in the infested palms; thus, to be really effective, treatments need to affect all different life stages, while in general the efficacy of pesticides including biopesticides is tested on neonate. Therefore, one of the objectives of the present study was aimed at knowing the efficacy of Bt through the determination of its median lethal concentration (LC_50_) and time (LT_50_) for the different stages of RPW.

At the level of insect immunity, a combination of the level of parasite threat, developmental stage, and sex may influence how much investment individuals will allocate to their immune system and the type of response [17,18]. Insects defend themselves against bacteria with humoral and cellular immune responses that have been shown to be an important determinant for stopping the progress of infections [19,20,21]. Recently, it has even been observed that insects display a phenomenon called “immune priming”, resulting in increased resistance upon repeated infection of an individual (within the generation) or of the progeny of immune-challenged parent(s) (across generations) [18]. However, research on how insect immunity changes with age, within an instar, or in larval stages is limited and contradictory [18,22,23]. For example, hemocytes play important roles throughout the growth and development of insects, and their response magnitudes can increase over ontogeny [1,24,25,26]. Moreover, hemocytes are the main component of cellular immune responses [1]. They are biochemically very sensitive towards change and have proven their ability to mediate diverse immune-defense processes against invading pathogens and other adverse conditions/stresses.

The number and the type of hemocytes (expressed as Total Hemocyte Count (THC) and Differential Hemocyte Count (DHC)) are basically influenced by environmental conditions such as humidity, temperature, photoperiod, disease, and stressors [27]; however, little is known about the interaction between entomopathogenic bacteria and hemocytes and its potential use as a detection method for testing pathogenicity.

In the RPW larval hemolymph, the most abundant hemocyte types are plasmatocytes (PLs) and granulocytes (GRs), which are involved in the clearance of bacteria and phagocytosis of other foreign organisms such as yeasts [4]. Indeed, larvae treated with Bt had a lower number of PLs than larvae fed with an untreated diet. Moreover, hemocytes and the (prophenol-) phenoloxidase system constitute the instantaneous innate immune system in RPW and other insects [5,9,28].

Infection by a pathogen induces biotic stress in the insect, stimulating the immune response [1,4,21]. These stressful situations stimulate organisms to undergo metabolic changes to adjust to new conditions, such as a modulation in the expression of stress proteins, as reported by Zhang [29], who highlight the importance of stress responses in the host’s defense systems. Of particular interest are proteins that are synthetized in all the cells of all living beings studied to date, from bacteria to humans, in response to the upper sublethal temperatures. Heat shock proteins (Hsps) were first discovered in the cells of the representatives from the class of insects, i.e., *Drosophila melanogaster* Meigen, 1830 (Diptera: Drosophilidae) [30]. Insects respond to elevated temperature and to a variety of chemical and physical stresses by rapidly increasing the synthesis of a set of conserved polypeptides collectively referred to as Hsps. They are a class of functionally related proteins involved in the folding and unfolding of other proteins and are named according to their molecular weight, such as Hsp100, Hsp90, Hsp70, Hsp60, Hsp40, small Hsp (sHsp), and Hsp10.

The induction and accumulation of HSPs have been studied extensively in response to various pollutants and pesticides [31,32] reported that the expression of Hsp70 in the midge, *Chironomus yoshimatsui* Martin and Sublette, 1972 (Diptera: Chironomidae), is induced in response to organophosphate (fenitrothion) and pyrethroid (ethofenprox). The authors suggested that Hsp70 is a sensitive indicator of low-level exposure to certain insecticides. However, studies tracing the effects of biopesticides on Hsp70 modulation in insects is scant, at least to our knowledge, and the biological role of Hsp70 and 90 remains to be elucidated [33,34].

One study has shown that baculoviruses serve as a stress factor that can activate both death-inducing and cytoprotective pathways in infected cells [35]. The infection potentiated the response to heat shock by boosting the Hsps contained in infected cells several-fold in comparison with uninfected cells [35]. The addition of a known inhibitor of inducible Hsps decreased the rate of viral DNA synthesis in infected cells and markedly suppressed the release of budded viruses, indicating the importance of the heat shock response for baculovirus replication [35].

Cross-protection occurred in the wax moth, *Galleria mellonella* L. (Lepidoptera: Pyralidae), when larvae were exposed to mild heat-shock at 38 °C, showing an enhanced humoral immune response after microbial infection in comparison to infected animals grown at 28 °C, and this was correlated with changes in the Hsp90 protein and increased level of the 55 kDa protein, suggesting that Hsp90 may play a significant role in converging pathways involved in insect immune response and heat-shock [36]. Previously, we showed for the first time that Hsp70 in hemocytes is modulated in RPW larvae after ingesting sublethal doses of Bt [37].

As in vertebrates, in insects Hsp expression and activity is also characteristic of nervous tissue playing a role not just in the maintenance and defense of cellular viability, but also in the preservation of neuron-specific luxury functions, particularly those that support synaptic activity. In fact, the modulation of HSPs is induced during stress and diapause in the insect brain [38].

In this study, we evaluate the effects of Bt on larvae, males, and females, considering not only mortality but also evaluating the immune system response. A new way to ascertain the pathogenicity of the effectors was assessed using hemocytes and brain tissue, and the role of Hsp70 in the biochemical responses in Bt stressed animals was evaluated.

## 2. Materials and Methods

### 2.1. Insect Collection and Rearing

Larvae and adults of *Rhynchophorus ferrugineus* (RPW) were collected from infested *Phoenix canariensis* in the province of Palermo (Italy, Figure 1) which had been cut down following phytosanitary measures for the control and eradication of RPW; specimens were collected in accordance with specific policy rules (Regional Decree 6 March 2007) and in collaboration with the Regional Phytosanitary Service (Unità Operativa 43 Osservatorio per le Malattie delle Piante di Palermo) and the Regional Forestry Agency (Azienda Regionale Foreste Demaniali) of the Region of Sicily.

In transit, the beetles were placed in plastic boxes containing palm tissue as a food source. The larvae and adult males and females were reared individually in Petri dishes (60 mm Ø) placed in a climatic chamber at 27 °C ± 0.9, 75% relative humidity, and a 24 h night photoperiod. RPW has sexual dimorphism: the male has a thicker rostrum with bristles. The animals were fed with pieces of apple (1.0 g ± 0.3). The development of the different larval instars was observed by detecting the molted head capsules. For the experiments, larvae at the later instar were used, which in our conditions corresponded to the seventh instar, characterized by high rate of feeding, high weight and high volume of hemolymph: (average weight 3.8 g ± 0.7, body length 3 cm ± 0.6, head capsule 6.0 mm ± 0.8). To carry out this research a total of 900 individuals were used.

### 2.2. Insect Bioassay: Lethal Concentration and Lethal Time

A commercial product consisting of 26,000 UI L.d./mg of Bt var. *kurstaki* Berliner pathotype H-3A, 3b EG 2424, registered against *Leptinotarsa decemlineata* (Say) (Coleoptera: Chrysomelidae), was used for the bioassay. According to the technical schedule provided by the producer (Intrachem Bio, Grassobbio, Italy), this strain (EG 2424) is a trans-conjugant strain with the genes of two different Bt strains, expressing extremely active crystal proteins against coleopteran pests.

To determine the median lethal concentration (LC_50_) required for Bt to kill 50% of RPW, larvae (as above described) and adults were fed with a diet containing 10 different concentrations: 0.5 mg/L, 1.0 mg/L; 1.5 mg/L, 2.0 mg/L, 2.5 mg/L, 3.0 mg/L, 3.5 mg/L, 4.0 mg/L, 5.0 mg/mL, and 6.0 mg/mL. There were 3 replicates of 10 individuals (each individual in a single Petri dish) for a total of 30 individuals for each concentration and for each instar, in addition to an untreated control. The control received only diet plus water. Individuals had been starved for 4 h before being used for the treatment. The Petri dishes were placed in a climate chamber at 27 ± 0.9 °C, 75% RH, and a 24 h dark (D) photoperiod. Mortality was recorded daily for 9 days.

To determine the median lethal time (LT_50_) required for Bt to kill 50 % of the RPW larvae and adults, a dose of 3.70 mg/L (almost corresponding to the LC_90_ calculated in the previous bioassay) was used to treat 30 RPW individuals for each instar. Specimens were singularly placed in Petri dishes with a small piece of apple, for each time interval. Similarly, ten larvae placed singularly in Petri dishes served as an untreated control and received only water and a small piece of apple. The mortality data was recorded after 3, 6, 12, 24, 48, 72, 96, 120, 144, 168 h, respectively. Mortality in the control was also noted to correct the mortality data according to Abbot’s formula, (Efficacy (%) = (1 − *n* in T after treatment/*n* in Co after treatment) × 100; where: *n* = insect population; T = treated; Co = control) [39].

### 2.3. Bleeding Procedure

Hemolymph was collected from each *R. ferrugineus* and kept separately to avoid allo-recognition reactions. The larval surface was sterilized with 70% ethanol for a few seconds and rinsed with sterile water and then dried. The specimens were anesthetized by placing them at −20 °C for 7 min, and hemolymph was withdrawn from the dorsal blood vessel with a sterile glass Pasteur pipet. About 1 mL hemolymph from each larva was collected in a sterile Eppendorf tube in the presence (*v*/*v*) of anticoagulant solution (98 mM NaOH, 186 mM NaCl, 17 mM Na_2_ EDTA, 41 mM citric acid and 10 mM phenylthiourea pH 4.5). To carry out the in vitro tests and hemocyte slide preparation, the hemolymph was centrifuged at 1000× *g* for 10 min. at 4 °C, and the pellet was suspended in 1 × 10^6^ cells/mL phosphate saline buffer (PBS). Adults were bled after the removal of the cephalic rostrum. About 200 µL of hemolymph were collected and immediately diluted in an equal volume of anticoagulant solution.

### 2.4. Bacillus Thuringiensis Effects on Hemocyte Number and Type

Time and dose were chosen in accordance with the results of the LC_50_ and LT_50_ bioassay in order to detect the stress response in earlier phases. Larvae and adults of both genders were fed with apple and treated with a 3.70 mg/L dose of Bt then taken at 0, 3, 6, 12, and 24 h to examine their hemolymph. Three replicates of 4 individuals for each stage, treated as described above, were considered for each of the 5 times. The hemolymph was collected and analyzed separately for each individual.

To count the hemocytes, immediately after bleeding pure hemolymph was put in a 0.0025 mm^2^ Neubauer improved hemocytometer (Assistant, Germany) under a light microscope (DLMB Leica, Wetzlar, Germany). Total hemocyte numbers (THC), expressed as ×10^6^/mL, and the differential hemocyte count (DHC) from untreated and treated Bt larvae, males, and females were recorded according to the type of hemocytes described by Manachini et al. [4].

### 2.5. Protein Extraction

#### 2.5.1. Hemocyte Lysate Supernatant

To prepare a hemocyte lysate supernatant (HLS) from RPW larvae and adults of both sexes, they were bled separately in the presence of anticoagulant (1 × 10^7^ cells/mL). After centrifugation at 1000× *g* for 10 min at 4 °C, the cell pellets were suspended in RIPA-buffer and sonicated (Sonifier Branson, model B-15 Danbury, CT, USA) for 1 min at 0 °C (1 pulse/s, 70% duty cycle) and centrifuged at 27,000× *g* for 30 min at 4 °C to remove any precipitate. Before use, the hemocyte extracts were dialyzed against 50 mM Trizma base (Tris(hydroxymethyl)aminomethane) pH 7.5, and the total protein content was determined using Bradford’s method [40]. To obtain a protein concentration sufficient to perform Western-blot analysis, the HLS from adults of three individuals were pooled.

#### 2.5.2. Brain Lysate Supernatant

After treatment with Bt, RPW male and female specimens and larvae were explanted the brain and a lysate supernatant (CLS) was prepared as follows: the tissue samples (20 mg ± 5) were powdered in liquid nitrogen with 1 mL of lysis buffer (RIPA: 0.5% sodium deoxycholate, minimum 97%; 1% NP40; 0.1% SDS with PBS-T (Na_2_HPO_4_ 1M, NaH_2_PO_4_ 1 M, NaCl 1.5 M, 0.1% Tween 20) pH 7.5, supplemented with a cocktail of protease inhibitors: 2 µg/µL antipain, leupeptin, and bestatin, 1 µg/µL aprotinin and pepstatin, 1 µM benzamidine, and 0.1 mM AEBSF. Then, the samples were centrifuged at 15,000× *g* for 30 min at 4 °C and the supernatant was collected. Before use, the extracts were dialyzed against 50 mM Trizma base (Tris(hydroxymethyl)aminomethane) pH 7.5, and the total protein content was determined by Bradford’s method [40].

### 2.6. Hsp 70 Estimation

The equivalent of 25 µg of total lysates for each sample was separated on 7.5% SDS-PAGE under reducing conditions according to Laemmli [41]. SDS-polyacrylamide minigels were transferred to nitrocellulose membranes using a semidry transfer apparatus [42]. To identify Hsp 70, the membrane was incubated with specific primary antibodies: anti-mouse Hsp 70 mAbs (Sigma-Aldrich, St. Louis, MI, USA) diluted in 3% BSA-TBS-T 1:2000 for HLS, and 1:3000 for CLS overnight at 4 °C. Then, the membrane was incubated with alkaline phosphatase-conjugated goat anti-mouse IgG 1:7500 for HLS and 1:10,000 for CLS. The integrated density value (IDV) was estimated by AlphaImager software (ProteinSimple, Orchard Parkway, San Jose, CA, USA).

### 2.7. Statistical Analyses

One-way analysis of variance (ANOVA) and the Tukey’s post-test were performed to find the significance between the concentration of Bt and percentage mortality with different instars. Differences among means were considered significant at a probability level of 5% (*p* ≤ 0.05). The concentration-mortality data were submitted to probit analysis to obtain a dose-response curve according to Finney [43] and then was calculated for the lethal times (LT_50_). The THC and DHC data were compared among treated and untreated individuals belonging to the same stage at different hours after treatment. Moreover, differences among means THC and DHC were analyzed in the control larvae and control females and males.

Each Hsp70 assay was performed in triplicate. Significant differences between values of by one-way ANOVA different incubation groups and the reference control groups were determined with statistical analysis and with Tukey’s post-test with SPSS Version 9.0.

## 3. Results

### 3.1. Laboratory Bioassay: Lethal Concentration and Lethal Time

The dose-response model provided a good fit to the data (*p* > 0.05), allowing the determination of toxicological endpoints, and confirms the toxicity of the Bt-strain to the different instars of RPW. Differences in RPW larval and adult mortality were detected among the treated individuals at different concentrations of Bt (Figure 2). No mortality was recorded in the control. Statistically significant differences were recorded in the susceptibility of the different instars for some concentrations (Figure 2). Females had the lower mortality compared to larvae for all doses except for the highest ones. Males were less susceptible than larvae but more than females except that for some doses (Figure 2).

Larvae were the most susceptible (LC_50_ = 1.22 mg/mL), followed by males, while females seemed to be more resistant (LC_50_ = 2.29 mg/mL), especially for the lower concentrations. These findings were confirmed by the LC_50_ and LT_50_ values and the results of the probit analysis (Table 1 and Table 2).

### 3.2. Effects of Bt on Hemocytes

Hemocyte numbers in the control larvae and males were significantly lower than hemocytes in females (Figure 3). The Bt treatment caused a dramatic decrease in THC in females at just after 3 h, in males after 6 h, and in larvae after 24 h (Figure 3). In treated individuals, the decrease in THC was also recorded for the rest of the experiment.

The differential hemocyte count (DHC) from RPW larvae and adults, carried out after Bt treatment, was compared to the controls, as shown in Table 3. Four types of hemocytes were recorded in all instars: plasmatocytes, granulocytes, oenocytes, and prohemocytes. Spherulocytes were recorded in larvae and in males. Three hours after Bt treatment, there was a decrease in all types of hemocyte populations. In particular, granulocytes, oenocytes, and prohemocytes had a statistically significant decrease in females, while plasmatocytes decreased significantly in males. This last type of hemocyte was statistically significant less than the control in larvae after 6 h from the treatment and in females after 12 h. In contrast, a significant increase in the number of oenocytes was recorded in larvae 12 h after Bt treatment.

### 3.3. Modulation of Hsp70 from RPW Larvae, Females, and Males in Hemocytes and the Brain after Bt Treatment

Western blot analyses using specific mouse monoclonal anti-Hsp70 antibody showed that the expression of the Hsp70 protein was modulated in response to Bt feeding (Figure 4A). The IDV values of Hsp70 (Figure 4B) in HLS increased in a statistically significant way to almost seven times compared to the control just 3 h after Bt treatment, while after 6 h it returned to the control value in all instars. Finally, it reached values similar to the control after one day (24 h) (Figure 4A,B). The expression levels of the Hsp70 protein in the brain of larvae, females, and males are shown in Figure 4.

In larvae, the IDV values revealed a rapid increase in expression after 3 h, with maximum expression at 6 h and remaining higher than the control for a whole day (Figure 5). The expression of the Hsp70 protein in the female and male brain had maximum expression at 6 h; it then rapidly decreased to control levels (Figure 5B).

## 4. Discussion

In Europe, *R. ferrugineus* was first detected at the beginning of the 1990s in southern Spain and, later, in 2005, it spread to Sicily where it has established itself. It is still one of the deadliest pests for palm trees worldwide, and the production of and trade in palm trees and related products are suffering, as are the landscape and natural and cultural heritage [16,44]. Insecticide applications are considered the most effective method for protecting palms from attack by palm weevils. Effective chemistries are varied: organophosphates, carbamates, neonicotinoids, and phenylpyrazoles. Accidental introduction of *Rhynchophorus* spp. into new areas has resulted in establishment and spread of these notorious palm pests in distinctly non-native habitats. Initial incursion detections are often in urban areas [45]. However, the recent data regarding the exposure of urban population to pesticides highlight the necessity of finding alternative control methods [46]. This could be even more urgent considering that studies have shown that several applications of chemical insecticides can increase the levels of enzymes which can lead to the development of resistance in *R. ferrugineus* [47]. In this contest *B. thuringiensis* is a valuable candidate. Although Bt products are the most sold bio-insecticides, there are still many interesting features to be investigated in the relationship of Bt and its hosts. We used the Bt-RPW interaction as a model to study the pathogen-host relationship. It is interesting that other insect pests of palms have been found to be susceptible to different Bt strains [48]. In particular, we wanted to investigate the stress induced by Bt infection on both larvae and female and male adults.

Initially, this paper describes bioassays that measured the mortality of later instar larvae, females, and males of *R. ferrugineus* exposed to varying dosages of Bt and exposure times to assess LC_50_ and LT_50_. The findings allowed the determination of a sub-dose to investigate the effect of Bt on the defense system of this pest. Results obtained with the proposed tests increased our knowledge about the potential pathogenicity of this entomopathogenic bacteria and highlighted the potential of this set of tests as a screening method to select new potential entomopathogens or bio-insecticides.

We considered the effect of Bt on mortality, estimating LC_50_ and LT_50_, hemocyte subsets, and stress response evaluated in the hemolymph and in the brain. The commercial Bt-based product had a negative influence on the mortality, immune system, and stress response of RPW larvae and adults of both sexes, although females showed less susceptibility. Thus, to control all stages would be useful to apply Bt considering the females susceptibility as in the infested palms are present in contemporary all insect instars. This study confirmed our earlier results [3,4,21] as Bt treatment resulted in a lower rate of survival.

The LC_50_ and LT_50_ found in this research are rather high, but it is important to notice that the product was not formulated against RPW and was used against the less susceptible instars so as to be able to detect immune and stress responses. In fact, when the doses were adjusted to the specimen’s weight, the levels of susceptibility were comparable. The lowest concentration, corresponding to approximately 130 (±7.0) µg/mL per fresh gram of larval weight and 170 (±5.0) µg/mL per fresh gram of adult’s weight, did not result in RPW mortality. The highest dose, corresponding to approximately 950 (±70.0) µg/mL per fresh gram of larval weight and 750 (±9.0) µg/mL per fresh gram of adult weight, produced 100% mortality. It is interesting to notice that, in our study, females were less affected and recovered more quickly than males and larvae, which was the most susceptible instar. In other studies, but with entomopathogenic nematodes, the most susceptible instar was the male [6].

It is proved that insect hematological parameters are affected by biotic factors, including pathogens, nematodes, and parasitoids, as well as abiotic factors, such as age, sex, pesticides, and starvation [17,18,19,20,26,27].

The literature says that our awareness of insect hemocytes and their functions in general is quite extensive. As a new trend, workers are now trying to use hemolymph as a medium for controlling insect pests because the changes occurring in the hemolymph are expected to be rapidly transferred to other portions of the body. Insect hemocytes represent a suitable cell type for analyzing differentiation, and their differential profile varies against stresses and other changes. Hence, hemocytes can be used as an indicator for change. Changes in THC and DHC in insects depend on their stress condition [27]. We have demonstrated that treatment with Bt decreased the THC values in both larvae and adults. Moreover, DHC also changed, emphasizing how the immune system acts in a different way according to the stage and the sex. According to Manachini et al. [4], plasmatocytes and granulocytes are the types more represented in RPW larvae. On the contrary, plasmatocytes are the second most prevalent hemocyte type in adults after granulocytes. They are reported to be involved in the encapsulation process and the phagocytosis of pathogens. Our results in the adults, after Bt treatment, showed their involvement in the response to infection. Cho et al. [49] found that non-entomopathogenic bacteria are actively engulfed by granulocytes between 6 and 12 h post-infection, and that granulocyte lysosomes were activated within 12 h to digest the bacteria, which were completely cleared by 48 h post-infection.

While in males the modulation of hemocytes fluctuated significantly, in females there was a drastic reduction, mainly in granulocytes and oenocytes. In the larvae, plasmatocytes and oenocytes were more affected by the Bt infection. In our particular case, the only hemocyte type that increased in Bt-treated animals were oenocytes in larvae.

In the literature, Makki et al. [50] hypothesized that, in *D. melanogaster* and *Tribolium castaneum* (Herbst) (Coleoptera: Tenebrionidae), oenocytes are involved in fatty acid and hydrocarbon metabolism. In particular, genetic studies have shown that larval oenocytes synthesize very-long-chain fatty acids and that adult oenocytes produce cuticular hydrocarbons required for desiccation resistance and pheromonal communication. This type cell may be useful for making energy available by acting on the fat body, a dynamic tissue involved in multiple metabolic functions. One of these functions is to store and release energy in response to the energy demands of the insect [51]. Moreover, the insect fat body plays a dual role in which it is both a metabolic organ, storing energy and providing energy to the rest of the organism, but also an organ important for humoral immunity [52]. In addition, the oenocytes are also involved in detoxification and response to infection [51,53]. Considering all of this, we can understand the trend of oenocity increasing in the larvae and decreasing in the females of *R. ferrugineus*, also keeping in mind that the production of cells in insects decreases as they pass from the larval to adult stage [53].

In our case, there was no variation in the number of granulocytes in RPW larvae that were also more susceptible than females to Bt, suggesting that granulocytes might give major resistance to the females also because the number of granulocytes in females is about 6 times higher than that of larvae. With regard to prohemocytes and spherulocytes, they are classified as non-phagocytic cells. In particular, spherulocytes are involved in the coagulation process and are fragile cells that break easily during sampling procedures [54]. Prohemocytes have features of precursor cells which are able to differentiate into other hemocyte categories and are the smallest cells in hemolymph [54,55]. Their role and modulation after Bt treatment in *R. ferrugineus* needs to be more thoroughly investigated.

This study, for the first time, examined the modulation of Hsp70 in the hemocytes and brain of larvae, females, and males of *R. ferrugineus* after an artificial diet treatment containing a sub-lethal dose of commercial Bt. The increase in Hsp70 protein expression reached its maximum at 3 h in hemocytes and 6 h in the brain in larvae, males, and females. Our data suggest that the effect of sublethal doses of Bt on *R. ferrugineus* may be mediated through the perturbation of cells as well as alterations in hemocyte composition and Hsp70 protein expression. It has been shown in the literature that, for protection against more acute forms of environmental stress, insects rely in part on the well-described heat shock proteins [56,57,58]. These proteins are rapidly up-regulated in response to such environmental insults as temperature extremes, anoxia, and various chemical contaminants. In particular, exposure to pathogens or parasitoids can induce the expression of Hsps. Larvae of *Heliothis virescens* (Fabricius) (Lepidoptera: Noctuidae) treated with baculovirus showed an increased expression of Hsp70 about 170-fold over control levels [59]. Larvae lethally infected with *Helicoverpa zea* (Boddie) (Lepidoptera: Noctuidae) single nucleopolyhedrovirus (HzSNPV) accumulated Hsp70 transcripts throughout the 72 h course of infection in the midgut, hemocytes, and fat body [60]. The expression level of Hsps in *Pieris rapae* L. (Lepidoptera: Pieridae) were influenced in response to parasitization [61].

In the brain of *Sarcophaga crassipalpis* Macquart (Diptera: Sarcophagidae), these proteins are expressed at very high levels and can persist for long periods during diapause or due to thermal shock [62,63].

Little is known about the relationship between Bt and Hsp expression in insects. Some studies have shown that Bt-Cry toxins can increase the expression of Hsp in the midgut of *Anthonomus grandis* Boheman (Coleoptera: Curculionidae) [64] which often bind to Cry proteins [65]. The Hsps protein in *G. mellonella* larvae infected with Bt showed changes in fat body Hsp90 levels after 90 min of Bt treatment [66].

Our results using pathogenic bacteria prompted us to conclude that the heat-shock protein Hsp70 may be an important part of the insect immune system at optimal growth temperature as its level increased following infection with *B. thuringiensis*.

## 5. Conclusions

In conclusion, our findings suggest that larvae appear to be the most susceptible target of Bt infection and that, in just a few hours, stress due to the infection can already be detected in the immune system and brain. Females were the less susceptible, and future studies must identify the precise factors mediating sex differences in the immune and stress responses, knowing that this will probably reflect complex interactions among hormones, genes, and our environment (both biotic and abiotic). Moreover, our integrated approach, comparing different potential targets of infection and different pathogenicity parameters (LD50, LT50, effect of pathogen on total number of hemocytes, ratio of selected hemocytes, expression of Hsp70 protein in the HLS) provides new insights into the evaluation of the most susceptible stages of pest with respect to this pathogen, and that it could be an alternative method for evaluating the most effective control agent that could be used to deal with this pest in the perspective of a sustainable Integrated Pest Management program.

## Figures and Tables

**Figure 1 insects-13-00437-f001:**
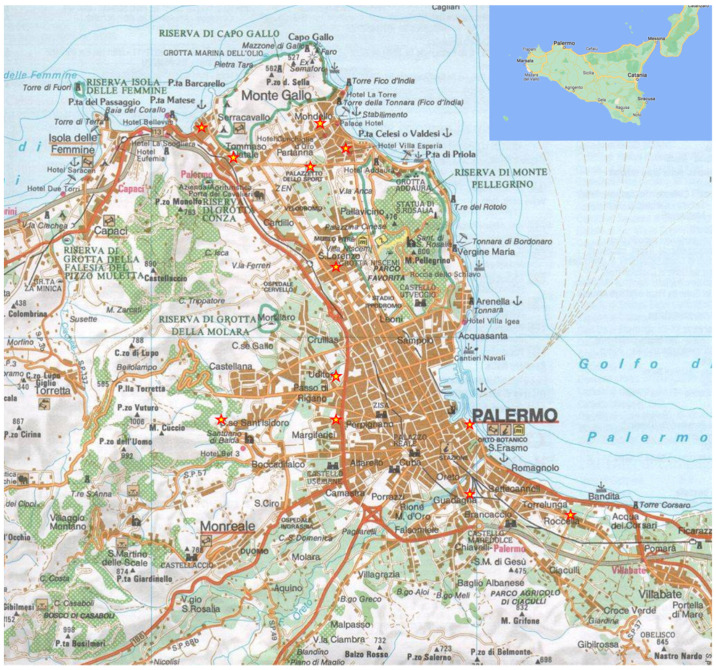
Map of study area. Symbols (yellow stars) indicate sampling localities.

**Figure 2 insects-13-00437-f002:**
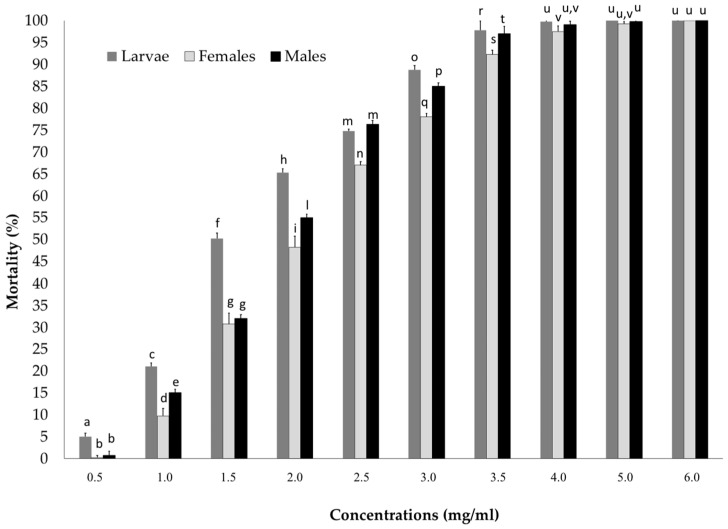
Mortality percentage of *Rhynchophorus ferrugineus* larvae, females, and males treated with *Bacillus thuringiensis* in a trial conducted at different lengths of time. Bars indicate standard deviations over 3 replicates of 10 individuals. Significant differences (*p* < 0.05) among the percentage of mortality are indicates with different letters.

**Figure 3 insects-13-00437-f003:**
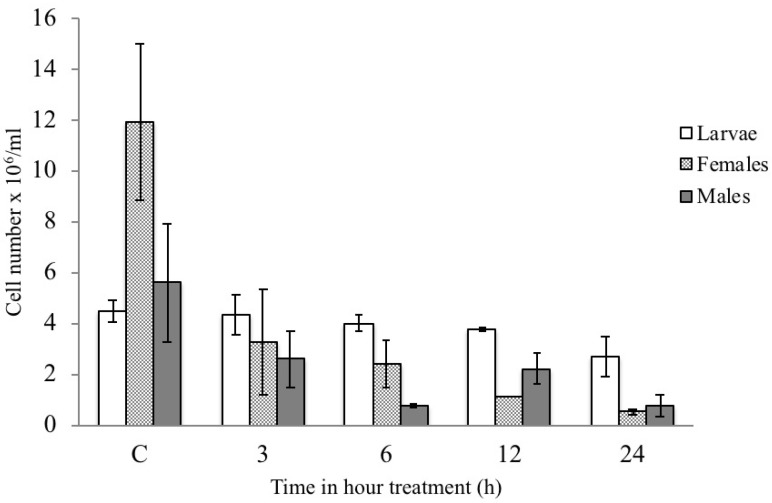
Mean (*n* = 10) and standard deviation of total hemocyte count (THC) recorded in the circulating hemolymph from RPW larvae, females, and males fed with untreated diet (control) and with diet containing Bt, collecting 3, 6, 12, and 24 h after treatment.

**Figure 4 insects-13-00437-f004:**
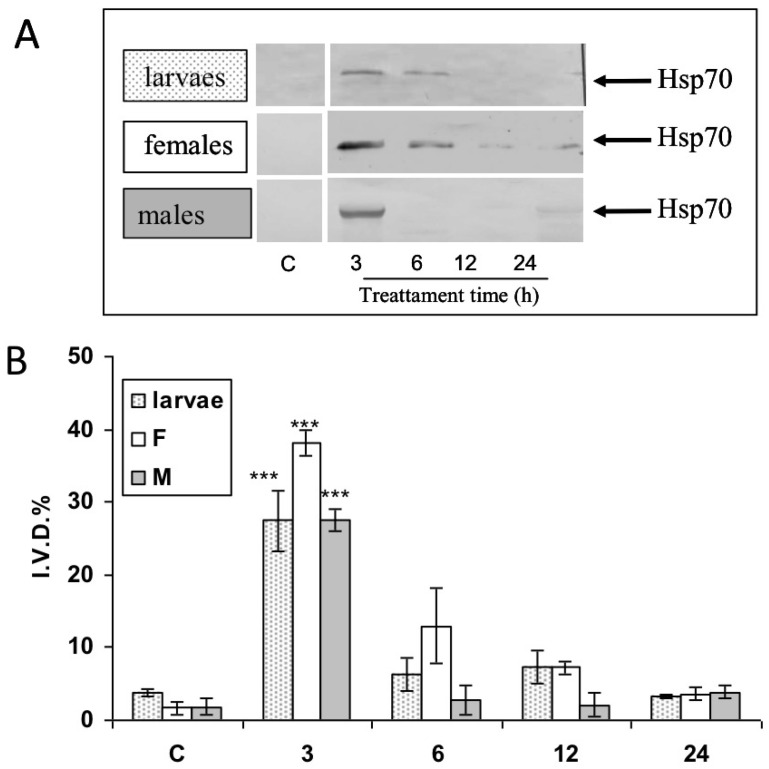
Effects of *Bacillus thuringiensis* on expression levels of the protein Hsp70 in adults and larvae of hemocites *Rhynchophorus ferrugineus* after 3, 6, 12, and 24 h treatment. (**A**) Representative Western blot of Hsp70 levels in three specimens from each group (one for each experimental trial). (**B**) Integrated optical density histogram (IDV) of the Hsp70 protein bands. Data are means ± S.D. (N = 15 control and N = 15 test specimens). Asterisks represent significant differences between control and test groups (*** *p* < 0.001).

**Figure 5 insects-13-00437-f005:**
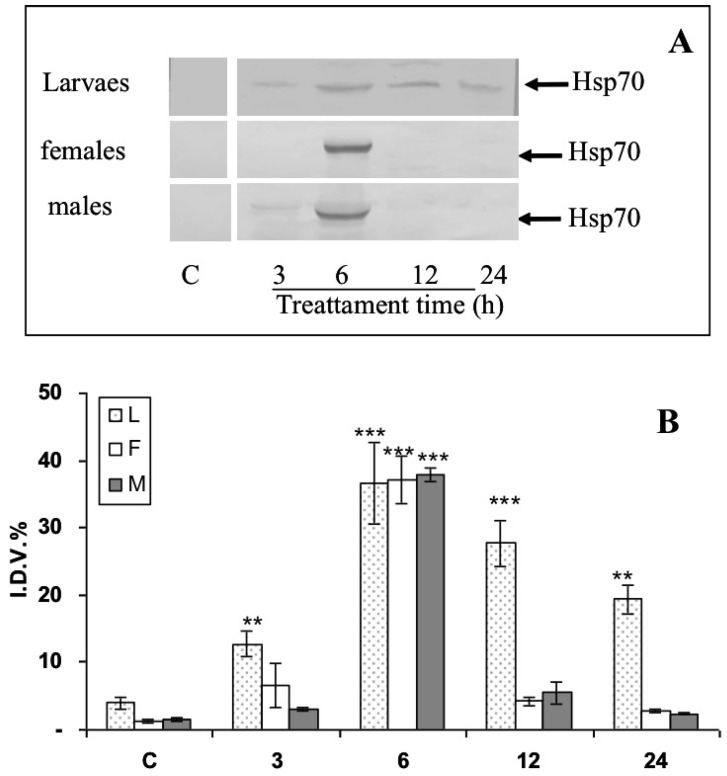
Effects of *Bacillus thuringiensis* on expression levels of the protein Hsp70 in adults and larvae of brain *Rhynchophorus ferrugineus* after 3, 6, 12, and 24 h treatment. (**A**) Representative Western blot of Hsp70 levels in three specimens from each group (one for each experimental trial). (**B**) Integrated optical density histogram (IDV) of the Hsp70 protein bands. Data are means ± S.D. (N = 15 control and N = 15 test specimens). Asterisks represent significant differences between control and test groups (** *p* < 0.01; *** *p* < 0.001).

**Table 1 insects-13-00437-t001:** Probit analysis of concentration-mortality of later instar larvae, females, and males of *Rhynchophorus ferrugineus* to *Bacillus thuringiensis*. NoI: number of insects; LC: lethal concentration; EC: estimated concentration; LC50: median lethal concentration (mg/mL); LC90: lethal concentration for 90% mortality; CI: confidence interval (mg/mL).

	No. I	LC	EC	95% CI	Slope ± SE	X^2^ (*p*-Value)
Larva	330	LC50	1.22	0.23–3.18	2.03 ± 0.09	7.44 (0.10)
Females	330	LC50	2.29	1.89–4.92	1.99 ± 0.11	3.77 (0.44)
Males	330	LC50	1.85	0.78–7.49	2.35 ± 0.10	6.54 (0.16)

**Table 2 insects-13-00437-t002:** The toxicity probit analysis of median lethal time of *Bacillus thuringiensis* against later instar larvae, females, and males of *Rhynchophorus ferrugineus.* NoI: number of insects; LT50: Median Lethal Time (hours); CC: Correlation Coefficient; CI: confidence interval (hours).

	No. I	LT50	Intercept ± SE	Slope ± SE	CC	LT50	95% CI
Larvae	40	36.82	5.96 ± 0.09	3.80± 0.45	0.940	36.82	32.76–40.95
Females	40	44.06	4.90 ± 0.10	2.98 ± 0.39	0.963	44.06	38.63–50.47
Males	40	37.34	5.23 ± 0.11	3.28 ± 0.41	0.984	37.34	32.78–42.90

**Table 3 insects-13-00437-t003:** Differential hemocytes count (DHC) recorded in the circulating hemolymph from larvae (L), females (F) and males (M) of *Rhynchophorus ferrugineus* fed with untreated diet (C = 0 h) and with diet containing Bt, collecting 3, 6, 12 and 24 h after treatment. Data are means ± s.d. (*n* = 10 control and *n* = 10 test individuals). Significant differences between the control and treated groups are shown (* *p* < 0.05; ** *p* < 0.01; *** *p* < 0.001). Significant differences among the circulating hemolymph from larvae (L), females (F), and males (M) at the same times are indicated with different letters.

Times	0 h	3 h	6 h	12 h	24 h
**Plasmatocytes (×10^6^/mL)**					
L	2.35 ± 0.45 a	1.99 ± 0.78 a,k	1.19 ± 0.33 t *	0.78 ± 0.35 ** α	0.26 ± 0.04 ** ι
F	3.42 ± 1.64 a	1.53 ± 2.35 a,k	0.95 ± 0.83 a,t,u	0.3 ± 0.00 * α,β	0.18 ± 0.08 * ι
M	0.87 ± 0.25 b	0.37 ± 0.13 * l	0.28 ± 0.03 u *	0.5 ± 0.36 b,β	0.55 ± 0.3 b,ι
**Granulocytes (×10^6^/mL)**					
L	1.52 ± 0.44 c	1.53 ± 0.38 c,m	1.52 ± 0.33 c,v	1.55 ± 0.09 χ	1.64 ± 0.2 c
F	7.97 ± 1.70 d	1.7 ± 0.33 ** m	1.38 ± 1.05 ** m,v,z	0.8 ± 0.01 δ **	0.32 ± 0.13 ** λ
M	4.38 ± 2.3 e	2.12 ± 0.97 e,m	0.5 ± 0.05 * w	1.65 ± 0.83 e,χ	0.21 ± 0.10 * λ
**Oenocytes (×10^6^/mL)**					
L	0.19 ± 0.04 f	0.36 ± 0.17 f,n	0.59 ± 0.28 f,x	0.67 ± 0.12 ** δ	0.44 ± 0.09 * μ
F	0.15 ± 0.02 f	0.05 ± 0.01 *** o	0.07 ± 0.02 ** y	0.05 ± 0.02 *** ε	0.03 *** μ
M	0.23 ± 0.19 f	0.08 ± 0.02 f,o	0	0.08 ± 0.06 f,ε	0
**Prohemocytes (×10^6^/mL)**					
L	0.32 ± 0.14 g	0.23 ± 0.06 g,p	0.37 ± 0.09 g,y	0.44 ± 0.09 g,φ	0.22 ± 0.06 g
F	0.43 ± 0.05 g	0 ***	0.002 ± 0.000 *** z	0 *** γ	0 ***
M	0.05 ± 0.05 h	0.03 ± 0.02 b,q	0	0	0.03 ± 0.00 ν
**Spherulocytes (×10^6^/mL)**					
L	0.14 ± 0.08 i	0.23 ± 0.05 i,r	0.35 ± 0.19 i,r,	0.38 ± 0.08 * η	0.18 ± 0.08 i
F	0	0	0	0	0
M	0.09 ± 0.01 j	0.001 ± 0.000 j,s	0	0.0015 ± 0.000 j	0

## Data Availability

The data presented in this study are available in the article.

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
