# Peer review of "Does Bacillus thuringiensis Affect the Stress and Immune Responses of Rhynchophorus ferrugineus Larvae, Females, and Males in the Same Way?"

_insects, 2022, doi:10.3390/insects13050437_

Round 1
Reviewer 1 Report
Line 28 – revise English.
Line 36 – please use biological terms such as sex.
Introduction:
Please reduce the number of paragraphs to only six.
Line 50 – Please join this sentence to a paragraph.
Line 67 – idem.
Line 82 – idem.
Materials and methods:
A map or a table indicating the location and coordinates of sampling sites is needed.
Line 241 – indicate that: the response variable in all bioassays was the number of dead RPW expressed as a percentage. Because mortality is a binomial variable dependent on counts, it does not meet normality assumptions. For this reason an ArcSin transformation … after that and Analysis of variance (ANOVA) …
The ANOVA was one-way or two-way? Why? Please state it.
Results:
Line 264 – indicate maximal efficacy and relative potency.
Discussion:
Please reduce the 10 paragraphs to only six. It would improve the flow of the discussion. Avoid paragraphs with one or two sentences.
Conclusions:
What does this study suggest on the effect of bt on larvae, males and females?
Indicate that although, efficacy of chemical insecticides still is high to controlling RPW, the search for new insecticides has been limited, mainly those that are less harmful to the environment and human health.
What is the importance of the evaluation of insecticides of natural origin for the control of this type of pests?
Author Response
Dear reviewer of “Insects”,
We revised the manuscript titled” Does Bacillus thuringiensis affect the stress and immune responses of Rhynchophorus ferrugineus larvae, females, and males in the same way?" according to your useful suggestions and corrections
We accept all your comments and we think that thanks to their contributions the manuscript is improved a lot. The only thing that we could not do a lot is to reduce the paragraph as the other two reviewers asked some changes and more detail in the introduction and discussion.
Below we respond in detail to the comments. We are also providing a revised manuscript that reflects your suggestions and comments.We feel that thsnk to your comments this manuscript has resulted in a stronger manuscript.
Line 28 – revise English. ANSWER - Done
Line 36 – please use biological terms such as sex. ANSWER - Done
Introduction:
Please reduce the number of paragraphs to only six. ANSWER - Only partially for the reasons explain above
Line 50 – Please join this sentence to a paragraph. Line 67 – idem. ANSWER - Only partially for the reasons explain above
Line 82 – idem.
Materials and methods:
A map or a table indicating the location and coordinates of sampling sites is needed. ANSWER – Done, thank you for the useful suggestion.
Line 241 – indicate that: the response variable in all bioassays was the number of dead RPW expressed as a percentage. Because mortality is a binomial variable dependent on counts, it does not meet normality assumptions. For this reason an ArcSin transformation ... after that and Analysis of variance (ANOVA) ... ANSWER - Done
The ANOVA was one-way or two-way? Why? Please state it. - ANSWER – Done – One-way
Results:
Line 264 – indicate maximal efficacy and relative potency. Answer - done
Discussion:
Please reduce the 10 paragraphs to only six. It would improve the flow of the discussion. Avoid paragraphs with one or two sentences. ANSWER - Only partially for the reasons explain above
Conclusions:
What does this study suggest on the effect of bt on larvae, males and females?
Indicate that although, efficacy of chemical insecticides still is high to controlling RPW, the search for new insecticides has been limited, mainly those that are less harmful to the environment and human health.
What is the importance of the evaluation of insecticides of natural origin for the control of this type of pests?
ANSWER – We add a full sentence to include your request.

Reviewer 2 Report
Interesting article on the potential use of B. thuringiensis in the management of R. ferrugineus in order to avoid the indiscriminate use of synthetic fungicides. Here are a few comments for your consideration.
Abstract: put the 50 of LD50 and LT50 in subscript
Line 26: add the abbrevation for Heat Shock Protein 70
Line 57: Please explain the meaning of Cry and Cyt toxins
Line 56: put the 50 of LD50 and LT50 in subscript. Review throughout the document (line 166, line 175, line 176, line 197, 245, 269,…)
Line 170: on this line the initials HSPs appear in uppercase and on other occasions in lowercase. Please put in the same format
Line 182: put Abbot's formula
Line 189: (v/v) in italics
Line 192: add the between cells and mL-1
Line 196: change Bt to the full name of the bacterium
Line 207: explain what the Manachini et al. registration method consists of. (4)
Line 211: put the units in the same format throughout the document, for example in line 211 cell/mL appears; while in line 192 it appears as cells ml-1
Line 217: Typo in the Bradford reference. It says that it is reference 4 and corresponds to 40
Line 254: put the p value in the same format, lowercase or uppercase (see line 243 and 286).
Line 260 (figure 1): change the coma of the concentrations (X axis) by points so that there are no errors (see line 168)
Author Response
Dear reviewer of “Insects”,
We revised the manuscript titled” Does Bacillus thuringiensis affect the stress and immune responses of Rhynchophorus ferrugineus larvae, females, and males in the same way?" according to your useful suggestions and corrections. Thank you to have spotted our mistakes.
We accept all your comments and we think that thanks to your contribution the manuscript is improved a lot. We are also providing a revised manuscript that reflects your suggestions and comments. We feel that thank to your comments this manuscript has resulted in a stronger manuscript.
Abstract: put the 50 of LD50 and LT50 in subscript ANSWER - Done
Line 26: add the abbrevation for Heat Shock Protein 70 ANSWER - Done
Line 57: Please explain the meaning of Cry and Cyt toxins ANSWER - Done
Line 56: put the 50 of LD50 and LT50 in subscript. Review throughout the document (line 166, line 175, line 176, line 197, 245, 269,…) ANSWER - Done
Line 170: on this line the initials HSPs appear in uppercase and on other occasions in lowercase. Please put in the same format ANSWER - Done
Line 182: put Abbot's formula. ANSWER - Done
Line 189: (v/v) in italics. ANSWER - Done
Line 192: add the between cells and mL-1
Line 196: change Bt to the full name of the bacterium ANSWER - Done
Line 207: explain what the Manachini et al. registration method consists of. (4) ANSWER - Done
Line 211: put the units in the same format throughout the document, for example in line 211 cell/mL appears; while in line 192 it appears as cells ml-1. ANSWER - Done
Line 217: Typo in the Bradford reference. It says that it is reference 4 and corresponds to 40 ANSWER - Done
Line 254: put the p value in the same format, lowercase or uppercase (see line 243 and 286). ANSWER - Done
Line 260 (figure 1): change the coma of the concentrations (X axis) by points so that there are no errors (see line 168) ANSWER - Done
Thank you,
Yours sincerely
Barbara Manachini
Reviewer 3 Report
These are my main comments on the MS (insects-1703609) by Monica Celi and colleagues.
The introduction and discussion provide no insight on how this MS relates to the various other ones cited in the text or concerns that have been raised by other researchers. This article should provide details on all these fronts to provide the proper context for the work. Authors do not present any hypotheses or expectations that could be connected to previous studies; adding these details will improve the paper. The authors should clearly explain WHY THE RESEARCH WAS DONE, WHY IT WAS IMPORTANT, and HOW IT FITS WITH OTHER STUDIES. It should be clear and concise. The introduction is too long and unstructured as is. The discussion should also include what outcome(s) they expect, and how it would help support or refute their hypotheses or answer their questions. Some of the authors statement would be much stronger if they tie their work to the body of literature that has built up on the invasion ecology and impacts of this pest in urban areas (see Journal of Pest Science, 2019 92(1), 143-156), as per authors’ discussion on invasiveness and impacts of this pest in urban environments throughout the text.
The M&M section is in poor shape, and statistical analysis specifically. E.g., ANOVA on percentage mortality data? If so, consider an analysis better suited to this response variable, logistic GLM is one, binomial distribution is another one (see Ecology 2011 92, 3-10). Results and Discussion section should be revised accordingly.
The discussion and conclusions lack real concluding remarks in my opinion, and if I was a practitioner or consultant, I’d want to see these recommendations for my area or city. How applicable these findings are to the real world? What are the true benefits of these findings? How would they fit into an existing IPM programs? Adding this information would benefit the discussion.
My other objections are about the language. I had difficulty understanding the English at times, which greatly affected my ability to understand whether the authors have entirely interpreted their results appropriately. There are quite a lot of errors, i.e., awkward phrasing, poor vocabulary, syntax and grammar errors, etc.
I was excited to see the results of the paper after reading the abstract, but I found it hard to extract key messages useful to policymakers and professionals, probably in large part due to the lack of connection with other published work and need for improved structure of the current manuscript. This is not to diminish the data gathered in this study, they are of value. The paper would benefit from a more thorough literature review and a better connection with more relevant reports on the subject.
The next draft of this paper will need to be dramatically different to have a chance at publication in my humble opinion.
Author Response
Dear reviewer of “Insects”,
We revised the manuscript titled ”Does Bacillus thuringiensis affect the stress and immune responses of Rhynchophorus ferrugineus larvae, females, and males in the same way?" according to your useful suggestions and corrections.
We are sorry that you do not appreciated our manuscript. It has as major scope studying the interaction between host and pathogen with particular focus on Bt and RPW as model. In this contest, it should not to be a fully applied research but a more basic science with some applicative aspects. This is the reason why we submitted the manuscript to “Insect” and not to a Journal more dedicated to applied entomology, such as Pest Management or Journal of Pest Science. Based on the model studied Bt-RPW we proposed a set of analysis to add to the classical mortality (e.g. expressed as LC 50, LC90, LT50, LT90) to select pathogen. To understand and know better insect immune system and how it can defeat or not defeat invading disease caused by bacteria, it is paramount also to improve the next generation of biopesticides based on entomopathogens.
For that reason, the research was not addressed to practitioners or consultants, that, as you claim, could not see recommendations on how to control RPW in their area or city. However, this was not the goal of our paper and we do not think that this kind information is searched by this kind of stakeholders in scientific journals but more in journals dedicated to practical aspects at local level. The paper is more addressed to entomologists, zoologists, biologists that are involved in studies on biology, physiology, and pathology of insects. Regarding the statistical analysis we thank you for pointing this out and giving us a chance to correct this oversight, as we wrongly wrote that we have done on percentage but was on real mortality. Thus, we correct the text.
Although we think that we revised a lot of papers on RPW, Bt, their relationship, insect immune system, RPW biology and control citing 64 papers, we understand that literature on that subjects is huge and we are sorry if we miss some important papers. On this regard, we want to thank you to point out the problem of RPW in the urban areas and we really appreciate the useful suggestion of literature that we have included in our manuscript. The problem of invasion of RPW and its presence in urban areas is really relevant and we humbly hope that our results could in future help to contrast this or similar pest in a more efficient way using also biological control agents.
Regarding English, please provide detail for what you think a poor English and where there are mistakes. Sorry for asking you that but because we paid a service and a translator for reviewing English, we really need your help to spot the mistakes and to complain with them. Thank you for understanding.
We hope that with the improvement also suggested by the other reviewers you can better consider our paper that want to be just another step on the basic knowledge of host-pathogen relationships with potential applied aspects and on the biology of this fascinating beetle.
Thank you,
Yours sincerely
Barbara Manachini
Round 2
Reviewer 3 Report
The authors have done a nice job addressing all my original comments and I thank them for it. Firstly, I truly appreciate their work as I have already noted in my original comments that the data are of true value and are publishable following revisions. I also appreciate that the authors have consulted English native speaker before submitting their manuscript to Insects. Some of the statements still sound/read odd to me, e.g., line_22: I’d advise using “against” or “for controlling the weevil” instead of “… potentially useful entomopathogen of the red palm weevil …”; line_24: I’d advise using the “pathogenicity” instead of “pathogenic activity”; line_61-65, this whole section needs to be clearer and more concise, e.g., the full name of this pest, both genus and species, has to be stated at the begging of each new sentence … also, where exactly is this pest ‘considered a major pest’ as described on Lines_61-62? … perhaps worldwide? … next, line_63 ‘… but several species of palms are also severely infested …’, I’d assume ‘… other palm species are attacked by this pest …’? … finally, Line_64 “thus to be really effective, treatments need to affect all different life stages.” reads completely off in my humble opinion, I’d assume that the treatments should target as many stages as possible because some of them are less or more susceptible to treatments??? … etc. etc. I’d leave it to the Editor to decide if these misspellings are acceptable by the journal.
The statistical analysis section needs more careful consideration before its publication, however. As already pointed out, the traditional transformations of proportion data were arcsine and probit. The arcsine transformation took care of the error distribution, while the probit transformation was used to linearize the relationship between percentage mortality and log dose in a bioassay. There is nothing wrong with these transformations, and they are available within SAS/R and other programs, but a simpler approach is often preferable, and is likely to produce a model that is easier to interpret. The major difficulty with modelling proportion data is that the responses are strictly bounded. There is no way that the percentage dying can be greater than 100% or less than 0%. But if we use simple techniques such as regression or analysis of covariance, then the fitted model could quite easily predict negative values or values greater than 100%, especially if the variance was high and many of the data were close to 0 or close to 100%. The logistic curve is commonly used to describe data on proportions, because, unlike the straight-line model, it asymptotes at 0 and 1 so that negative proportions and responses of more than 100% cannot be predicted. Briefly, proportions are based on number of cases. Would you give the same weight to a proportion of 2 out of 4 cases (not very reliable) and a more reliable proportion of 20 out of 40 cases? The natural solution is to use the odds and odds ratio, and a binomial distribution to test for change in proportion as a change in the odds, as described in the arcsine asinine publication. That way you give 50 % of 40 its due, compared to 50% of 4.
The authors are now stating that the arcsine and probit were used on non-binomial ‘real mortality data’ instead of on ‘percentages’ which is erroneous. For non-binomial data such as ‘count mortality data’, the arcsine transform is ‘undesirable on the grounds of interpretability, and because it can produce nonsensical predictions’, as described in the arcsine asinine publication. Where data are non-binomial, there is no motivation to use the arcsine transform at all, and instead efforts should be placed into searching for a transform that satisfies linearity assumptions, while if possible, being useful for interpretation. If data are non-binomial, i.e., not of the form “x out of n,” then logistic regression is no longer applicable as suggested for percentages, and usually the distribution of the data is no longer known. The logit transformation is proposed as an alternative approach to address these issues. Consequently, there is no justification to prefer the arcsine transform over any other transform, as variance stabilization is now no longer a goal. Hope this clarifies it a bit and please revise accordingly.
Thank you and good luck!
Author Response
Dear Reviewer,
we are grateful for your insightful comments on our paper. We have been able to incorporate changes to reflect most of the suggestions provided by you. We have highlighted the changes in light blue within the manuscript.
Thank you for pointing this out. We agree fully with these your comments: “Some of the statements still sound/read odd to me, e.g., line_22: …..line 61-62….line 64…”
Therefore, we have added the requested information and changed it according to your suggestions. In addition, we have done again the statistical analysis according to your suggestions. Revising the data, we were able to include the data of the two highest doses that we did not include before but we think can help more the readable. Thank to your suggestions and the opportunity to revise the statistical analysis, in consequence of that we have also changed the figure 2 which now is more informative
Sincerely,
Barbara Manachini